# ON THE PRACTICALITY OF BOLTZMANN NEURAL SAMPLERS

## ABSTRACT

We tackle a challenge at the heart of the missions of computational chemistry and biophysics—to sample a Boltzmann-type distribution

$$p(\mathbf{x}|\mathcal{G}) \propto e^{-U(\mathbf{x}|\mathcal{G})} \tag{1}$$

on $\mathbb{R}^{N \times 3}$ associated with some $N$-body system $\mathcal{G}$, where $U$ is an energy function (termed *force field*) with orthogonal invariance and *deep, isolated* minima. Traditionally, this is sampled sequentially using Markov chain Monte Carlo methods, which can be so slow that one, for weeks of wall time, never breaks free from the local minima defined by the starting pose. Neural samplers have been designed to speed up this process by optimizing the dynamics, prescribed by a stochastic differential equation (SDE). Though sound and elegant in continuous time, they can be practically unstable and inefficient when discretized. In this paper, we attribute this phenomena to the limited expressiveness of the finite additive transition kernels, and their inability to bridge distant distributions. To remedy this, we design a new type of highly flexible prior by mixing orthogonally invariant densities (Mint), as well as a new discretized non-volume-preserving kernel, termed Jacobian-unpreserving Langevin with explicit projection (Julep). Together, MintJulep greatly improve the practical performance of neural samplers, while keeping the underlying SDE intact.

## 1 INTRODUCTION: BOLTZMANN DISTRIBUTION AND NEURAL SAMPLERS

Statistical mechanics, some [1] say, bridges the microscopic and the macroscopic world,

$$\bar{\mathcal{O}} = \int \mathrm{d}\mathbf{x}\mathcal{O}(\mathbf{x})p(\mathbf{x}), \tag{2}$$

with the probability distribution $p$, conditioned on some $N$-body system $\mathcal{G}$, adopting the Boltzmann [2] form (Equation 1), known up to a constant. On one end of the bridge are per-frame ($\mathbf{x} \in \mathbb{R}^{N \times 3}$) computable quantities $\mathcal{O}(\mathbf{x})$; on the other, $\bar{\mathcal{O}}$, some ensemble observable tangibly measurable in laboratories, such as the binding affinity of a newly designed therapeutics, or the physical properties of an innovative material. As such, to draw samples from Equation 1 in an efficient and unbiased manner to estimate Equation 2, will shed quantitative light on the understandings and discoveries spanning various domains, from chemistry, material science, to biophysics. Many machine learning pipelines in these disciplines can be seen as approximating (force field construction [3–10]) or minimizing (conformer generation [11], docking [12, 13], and protein folding [14–16]) the Boltzmann distribution. Nevertheless, if one wishes to rigorously sample such distribution till convergence, Monte Carlo methods are typically needed, known as molecular dynamics (MD) simulations [17–19], which is slow and biased towards the starting pose, due to the sequential nature.

**Preliminaries.** The aforementioned sampling process typically involves integrating a SDE (from $t = 0$ to $1$ without loss of generality), using, for instance the overdamped Langevin dynamics,

$$\mathrm{d}X = -\epsilon\nabla U_t \mathrm{d}t + \sqrt{2\epsilon}\mathrm{d}B, \quad X_{t=0} \sim q_0 \tag{3}$$

where $\epsilon$ denotes the volatility (inverse friction). $U_1 = U$ is required to target the correct Boltzmann distribution. A constant $U_{t \in [0,1]} = U$ with a static starting point $q_0 = \delta(\mathbf{x})$ represents the traditional

sampling process ubiquitously used by MD practitioners. Alternatively, a tractable starting point, such as a wide isotropic distribution $q_0 = \mathcal{N}(0, \sigma I), \sigma >> 0$, together with a linearly annealing potential $U_t = tU + (1-t)(-\log \mathcal{N})$, recovers the simulated annealing approach. When $U_t$ is not constant, the quantity

$$W(X) = \int_0^1 \mathrm{d}t \partial_t U_t(X), \tag{4}$$

known as *generalized work* for physicists and *path weights* for statisticians, can be used as to correct for the bias introduced in this process when estimating arbitrary functions $f$ (annealed importance sampling, AIS [20]), as well as to estimate the ratio of normalizing constants between $k_0$ and $k_1$ (Jarzynski's equality [21], JE):

$$\int \mathrm{d}X f(X) k_1(X) = \frac{\mathbb{E}[e^W f(X_1)]}{\mathbb{E}[e^W]}; \frac{Z_1}{Z_0} = \frac{\int \mathrm{d}X_1 k_1(X_1)}{\int \mathrm{d}X_0 k_0(X_0)} = \mathbb{E}[e^W]. \tag{5}$$

**Problem statement & related works.** To improve the convergence of $X_1$ to Equation 1, a non-equilibrium control [22] $b_t$ can be added to the drift term in Equation 3, resulting in the *nonequilibrium annealing process*:

$$\mathrm{d}\overrightarrow{X} = -\epsilon\nabla U_t \mathrm{d}t + \sqrt{2\epsilon}\mathrm{d}B + b_t \mathrm{d}t, \tag{6}$$

with the corresponding path weights:

$$W(X) = \int_0^1 \mathrm{d}t(-\nabla \cdot b_t(X) + \nabla U_t(X) \cdot b_t(X) + \partial_t U_t(X)). \tag{7}$$

A perfect control term [23] exists so that $X_0$ can be transported to exactly match $X_1$ to Equation 1, mitigating the need of reweighting, i.e., $W \equiv 0$. We call the neural parametrization and optimization of $b_t$ towards this goal *neural samplers*. For this purpose, the most obvious choice of objectives seeks to formulate this as a stochastic optimal control (SOC) problem [24–27] minimizing the control energy and the terminal reverse-KL divergence $D_{\mathrm{KL}}[q_1||k_1]$, where $q_1$ is governed by the law of Equation 6. These *online* approaches require the differentiation through the SDE integration, and can therefore be expensive or unstable. When $\epsilon = 0$, the deterministic counterpart of Equation 6 reduces to a (continuous) normalizing flow [28–30], referred to as Boltzmann generators [31, 32] in our context. To speed up convergence and prevent mode-collapsing ubiquitous in reverse-KL-based methods requiring only the energy function (*energy-based training*), these types of approaches typically requires samples from $k_1$ (*data-based training*) to evaluate the forward-KL $D_{\mathrm{KL}}[k_1||q_1]$, incurring an overhead. Overall, *offline* methods relying on neither samples nor differentiable trajectories seem theoretically attractive for scalability. Specifically, consider a backward SDE with $X_0 \sim k_1$:

$$\mathrm{d}\overleftarrow{X} = -\epsilon\nabla U_t \mathrm{d}t + \sqrt{2\epsilon}\mathrm{d}\overleftarrow{B} - b_t \mathrm{d}t. \tag{8}$$

With Equation 7, the *controlled* [33] Crook's fluctuation theorem reads exactly like (and recovers, when $b \equiv 0$), the original Crook's fluctuation theorem (CLT [34]):

$$\mathrm{d}\overrightarrow{\mathbb{P}}/\mathrm{d}\overleftarrow{\mathbb{P}} = \exp(W - Z_0 + Z_1), \tag{9}$$

where the derivative is taken in the Radon-Nikodym (RND) sense, and $\overrightarrow{\mathbb{P}}, \overleftarrow{\mathbb{P}}$ are the path measure associated with Equation 6, 8, respectively. This furthermore generalizes JE (5) since $\mathbb{E}[\mathrm{d}\overrightarrow{\mathbb{P}}/\mathrm{d}\overleftarrow{\mathbb{P}}] = 1$. Although many divergences can be employed to find the perfect control term [35], such as the physics-inspired neural network (PINN [36]) or the action matching (AM [37]) loss, the most straightforward offline method [33, 38] minimizes the log-variance of the RND (9):

$$\mathcal{L} = \mathbb{V}[\log[\mathrm{d}\overrightarrow{\mathbb{P}}/\mathrm{d}\overleftarrow{\mathbb{P}}]], \tag{10}$$

with can be taken w.r.t. any measure (hence the offline nature), albeit usually w.r.t. $\overrightarrow{\mathbb{P}}$ for minimal variance. When this approaches zero, $b_t$ is perfect since $W \equiv 0$ and $\overleftarrow{\mathbb{P}}$ is exactly the time reversal [39] of $\overrightarrow{\mathbb{P}}$.

**Pathology: Why neural samplers fail in practice?**    While the aforementioned formulation is simple and elegant in theory (continuous-time), practically, when discretized, it fails to perform when realistic physical systems are involved [26]. We postulate that this can be attributed to the limited expressiveness of the discretized kernel, and its inability to bridge drastically distant distributions. Slightly formally, if we do not consider the equilibrium part of the drift $\nabla U$ (corresponding to the infinite-inertia scenario in physics), we observe that (proof in the Appendix):

*Remark* 1.1 (Expressiveness of local kernels.). Let $(k_t)_{t=0}^{T-1}$ be Markov kernels of the form

$$k_+(X_{t+\Delta t}|X_t) = \mathcal{N}(X_t + b_t(X_t)\Delta t, \sqrt{2\epsilon}I) \tag{11}$$

where each drift map $f_t(x) = x + b_t(x)\Delta t$ is $L$-Lipschitz. Let $q_0$ be the initial law and define $q_T := q_0 k_0 k_1 \cdots k_{T-1}$. Assume there exists a reference measure $\mu$ such that $W_1(q_0, \mu) \leq 1$  and  $W_1(q_1, \mu) = W_1$, for some target distribution $q_1$ with (normalized) 1-Wasserstein distance $W_1 \gg 1$ from $\mu$. If $q_T = q_1$, then necessarily $T \geq \log_L W_1$. In words, at least $\log_L W_1$ discrete kernels are needed to transport $q_0$ to $q_1$ using such local $L$-Lipschitz steps.

**Main contributions.**    The aforementioned pathology tells us that the practical underwhelming performance of neural samplers can be attributed to (1) the gap between the target distribution $p$ and the *tractable* distribution $q_0$ at $t = 0$, henceforth referred to as the *prior* of neural sampling, and the target distribution $p$; and (2) the inability for additive kernels in the form of Equation 11 to gap such gaps. Motivated by this, while keeping the SDE (Equations 6, 8) and the objective (Equation 9) intact, we propose:

- A new prior called **Mint** (mixture of invariant densities, §2), where we achieve high parametrized flexibility while respecting the symmetry of physical systems.
- A new discretized kernel termed **Julep** (Jacobian-unpreserving Langevin with explicit projection, §3), which adds additional expressiveness to each step by allowing not only additive but also multiplicative transformations.

Further relating the discoveries to prior literature, we note that [40] also optimizes the prior of the sequential Monte Carlo process while leaving the actual annealing dynamics invariant, albeit using a much more detailed but expensive invertible flow model evaluated only once during the SDE integration. Blessing et al. [41] also proposes Gaussian mixtures as the initial distribution of the SDE of the diffusion process; the Mint prior can be seen as the orthogonally equivariant version of this idea. The Julep kernel, on the other hand, can be regarded as a continuous stochastic normalizing flow model [27] sandwiched by deterministic bijections built with matrix exponential [42], of which the time-discretized integration on a graph manifold is inspired by [43]. In §4, we show that these two innovations greatly enhance the feasibility of Boltzmann neural samplers, bringing us one step closer to the efficient and scalable sampling of Boltzmann distributions for physical systems.

## 2    MINT PRIOR: MIXTURE OF ORTHOGONALLY INVARIANT DENSITIES

To model the highly irregular distributions defined by the force field $u$ in Equation 1, which we know very little except that it is orthogonally invariant (w.r.t. internal rotation or reflection, definition above), we ask the question: whether it is possible to find a class of distribution that can approximate any arbitrary distributions on $\mathbb{R}^{N \times n}$ up to the orthogonal symmetry group $O(n)$? Formally:

**Definition 2.1.** A function $f$ is said to be orthogonally invariant on $\mathbb{R}^{N \times n}$ if $\forall X \in \mathbb{R}^{N \times n}, Q \in \mathbb{R}^{d \times d}, QQ^\top = Q^\top Q = I$,

$$f(X) = f(XQ). \tag{12}$$

A distribution is said to be orthogonally invariant if its density function satisfies Equation 12.

It is also worth noting that, while most of the force fields used by computational physicists and chemists are actually $E(n)$-, rather than $O(n)$-invariant, we constrain the translational degrees of freedom here and work with a radial, internal coordinate system. Practically, this can be done by consistently placing one particle, termed *anchor atom* henceforth (See an illustration in Figure 2.), at the origin of the coordinate system. In addition, although we are more interested in $n = 3$, all results shown in this paper can be generalized to all $n \in Z^+$, as we do not reply on any operators other than

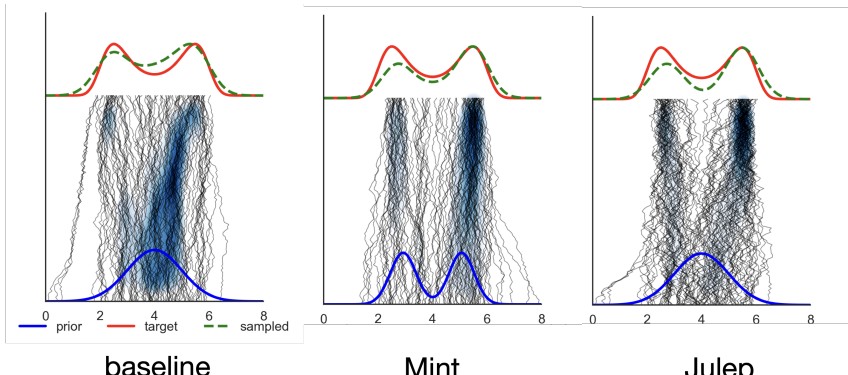

Figure 1: **Semantic illustration.** Sampling tarjectory of DW-2 with the DDS [25] baseline (left), Mint prior (middle), and Julep kernel (right). Mint accelerate the sampling by parametrizing a flexible kernel; Julep accelerate the sampling by employing non-local, cross-mode moves.

the dot product. In the following section, we firstly handle the angular degrees of freedom, before we proceed to the radial part and derive a parametric class of distribution capable of approximating any densities up to the defined symmetry.

**Pairwise von Mises-Fisher** (PvMF) **distribution:** $O(n)$-**invariant on** $(\mathbb{S}^{n-1})^N$**.** To start, we propose a new class of distribution termed the *pairwise von Mises-Fisher* distribution, on the manifold $(\mathbb{S}^{n-1})^N$, which is the $N$-product of $\mathbb{S}^{n-1}$ spheres:

**Definition 2.2** (PvMF distribution)**.**

$$\mathrm{PvMF}(\Theta; \mu, \kappa) \propto \exp(\kappa \cos\langle \mathrm{vec}(\Theta\Theta^\top), \mathrm{vec}(\mu\mu^\top)\rangle), \tag{13}$$

where $\kappa \in \mathbb{R}^+$ and $\Theta, \mu \in (\mathbb{S}^{n-1})^N$. Simply put, the energy function measures the *cosine similarity* between the gram matrices defined respectively by the variable $\Theta$ and the parameter $\mu$, both on the $(\mathbb{S}^{n-1})^N$ manifold, and prescribes the density similar to the vanilla von Mises-Fisher (vMF) distribution [44]. Since such a density peaks at the perfect alignment of $X$ and $\mu$ up to an orthogonal transformation $Q$, with $\mathrm{PvMF}(\mu Q, \mu) \propto \exp(\kappa) < \infty$, and therefore the integration over a finite-volume manifold $Z = \int_{\Theta \in (\mathbb{S}^d)^N} d\Theta \mathrm{PvMF}(\Theta, \mu) < \infty$ is normalizable. Besides, the gram operator is $O(n)$-invariant, so it naturally follows that:

*Remark* 2.3. $\mathrm{PvMF}(\Theta; \mu, \kappa)$ is a valid probability distribution.

*Remark* 2.4. $\mathrm{PvMF}(\Theta; \mu, \kappa) = \mathrm{PvMF}(\Theta Q; \mu, \kappa), \forall Q^\top Q = I$ is orthogonally invariant.

Evidently, this probability density requires $\mathcal{O}(N^2)$ runtime complexity to evaluate.

**Mixture of** PvMFLogNormal **products: universally approximative.** Having the angular degrees of freedom ($\Theta = X/\|X\| \in (\mathbb{S}^{n-1})^N$) taken care of, we pair the PvMF distribution with a simple LogNormal distribution on the radial axis ($r = \|X\| \in \mathbb{R}^N$) and can now define a distribution on the entire $X \in \mathbb{R}^{N \times d}$ space, called the PvMFLogNormal product, which stays orthogonally invariant:

$$\mathrm{PvMFLogNormal}(\Theta, r; \mu, \kappa, \rho, \sigma) = \mathrm{PvMF}(\Theta, \kappa)\mathrm{LogNormal}(r; \rho, \sigma), \tag{14}$$

where $\Theta \in (\mathbb{S}^{n-1})^N, r \in R^+$ and $\mu \in (\mathbb{S}^{n-1})^N, \kappa \in \mathbb{R}^+, \rho, \sigma \in \mathbb{R}^N$. We now arrive at the complete form of the family distribution used henceforth—the mixture of PvMFLogNormal products, followed by the expressive characterization.

**Definition 2.5** (Mixture definition)**.**

$$q(X; \{\pi_i, \mu_i, \kappa_i, \rho_i, \sigma_i\}) = \sum_i \pi_i \mathrm{PvMFLogNormal}(X/\|X\|, \|X\|; \mu, \kappa, \rho, \sigma), \tag{15}$$

**Theorem 2.6** (Universal approximator)**.** *Mixture of* PvMFLogNormal *distributions with sufficient components can approximate any arbitrary Riemann-integratable orthogonally invariant distributions on* $\mathbb{R}^{N \times d}$ *with arbitrarily small error.*

The proof, deferred to the Appendix, follows the first fundamental theorem of the orthogonal group [45] and the style of the universal approximation theorem of the Gaussian mixture models [46].

**Sampling and energy-based variational inference (VI).** While the family of distribution is defined, we realize that sampling from this distribution (or the PvMF distribution itself) is highly non-trivial. Recall that ([44] §3.5.22 and 9.3.15), for regular vMF distributions, in the high concentration limit ($\kappa \to \infty$), its behavior converges to that of a projected normal distribution, which is easy to sample. Following the same procedure to Taylor-expand the density on the tangent space $I - \mu\mu^\top$, we see that our PvMF distribution, when concentrated, can also be approximated by a projected normal distribution rotated by an arbitrary angle $Q$ (or reflection):

$$\mathrm{PvMF}(\Theta; \mu, \kappa) \approx \mathcal{PN}(\Theta; \mu Q, 1/\sqrt{\kappa}), \forall Q, \kappa \to \infty. \tag{16}$$

This approximation efficiently generates samples from the PvMF distribution, which can be further corrected by a brief Langevin dynamics integration. If we are working with a problem where the loss function is also orthogonally invariant, as is in this paper, the rotation can also be practically emitted $Q = I$. In this case, although the proposal distribution $\mathcal{PN}$ is not orthogonally equivariant, the resulting PvMF is. These samples are then used downstream to be multiplied by the radial components and blended into a mixture.

Samples from $q$ (Equation 15) at hand, we can easily fit this highly flexible function to arbitrary target densities $p$ by optimizing, for instance, the reverse KL divergence $D_{\mathrm{KL}}[q||p] = \mathbb{E}_q[\log q - \log p]$. Here, since we are dealing with particularly rugged energy landscape where the gradient of $p$ can be numerically overwhelming, we adopt the trick from [40] and optimize the REINFORCE [47] policy gradient surrogate instead:

$$\mathcal{L}_{\mathrm{Mint}} = -\mathbb{E}_q \log q[\overline{\mathrm{SoftMax}}(\log p - \log q)] \tag{17}$$

This objective fills the energy landscape with elliptical probability masses. The mode-seeking behavior of the reverse KL-divergence is not problematic here as the multimodal nature of $p$ can be captured by explicit discrete mixtures (which is one of the greatest challenges of neural samplers) [48]. We can also add an additional Stein VI [49]-style repulsion kernel among the gram matrix of the mixture components to encourage the diversification of modes.

$$\mathcal{L}_{\mathrm{Repulsion}} = \sum_i \sum_{j \neq i} \cos(\mu_i \mu_i^T, \mu_j \mu_j^T) \tag{18}$$

As such, we can view Mint as the *optimization* stage of neural sampler training, quickly and cheaply finding diverse minima on the energy landscape. In § 4, we see that Mint alone can achieve satisfactory results in terms of mode finding. Of course, despite of Theorem 2.6, in the finite limit of the number of mixtures, the elliptical density $q$ cannot fill arbitrarily sophisticated shapes. This motivates the design of a highly expressive kernel in the following section.

## 3 JULEP KERNEL: JACOBIAN-UNPRESERVING LANGEVIN WITH EXPLICIT PROJECTION

We design a novel forward kernel $k^+$ to replace that in Remark 1.1, together with its corresponding backward kernel $k^-$:

$$k_\pm(X_{t\pm\Delta t}|X_t) = \mathcal{N}\Big(\exp\big(\pm A_t(X_t)\Delta t\big)X_t \pm b_t(X_t)\Delta t + \partial U_t/\partial X \Delta t, 2\epsilon\Delta t\Big), \tag{19}$$

where $\exp$ denotes matrix exponential and $U = -\log p$ up to a constant. One can easily see that this is but a different discretization of Equation 6, now written as $\mathrm{d}\overrightarrow{X} = -\epsilon\nabla U_t \mathrm{d}t + \sqrt{2\epsilon}\mathrm{d}B + \big(b_t(X_t) + \exp W_t(A_t)\big)\mathrm{d}t$, with the last term omitted into $b_t$ by considering the first-order Taylor expansion of the matrix exponential. Intuitively, our method affords the traditionally additive kernel a multiplicative structure, thus greatly enhancing the expressiveness of each step, allowing it to bridge faraway distributions. These intuitions can be formalized as the following remarks, with proofs in the Appendix.

*Remark* 3.1 (Consistency of Julep kernels.). The discretized kernel $k$ as defined in Equation 19 is a *consistent* solution to the SDE Equation 6, i.e. it recovers the original SDE when $\Delta t \to 0$.

*Remark* 3.2 (Julep kernel has the same strong convergence as Euler-Maruyama.). The discretized kernel $k$ as defined in Equation 19 converges strongly to the SDE Equation 6 with order $1/2$. In other words, if $b, \sigma, \exp(A)$ are $L$-Lipschitz and satisfies the linear growth condition, if $X_t$ is a true solution to Equation 6 and $\hat{X}_t$ is a discretized solution, there exists a constant C, such that

$$\sup_{0 \leq t \leq 1} |X_t - \hat{X}_t| \leq C\sqrt{\Delta t} \tag{20}$$

*Remark* 3.3 (Julep kernel breaks the expressiveness bottleneck.). Following the setting in Remark 1.1, except for the kernel definition in Equation 19, and if $b_t \Delta t$ and $\exp(A_t)$ are $L$-Lipschitz, the number of discretized kernels needed to transport $q_0$ to $q_t$ is smaller than $\log_L W_1$.

Following [23, 27, 50, 51], the path weight (Equation 4) can be discretized as:

$$W \approx U_0(X_0) - U_1(X_1) + \sum_{t=0}^{1} \log k^+(X_t | X_{t-\Delta t}) - \sum_{t=0}^{1} \log k^-(X_{t-\Delta t} | X_t). \tag{21}$$

This relies on the assumption of the Gaussianality [33] of the reverse kernel, which is exact when $\Delta t \to 0$. Plugging this into the log-variance objective (Equations 9, 10) and omitting the constant $Z_0 - Z_1$, we arrive at the log-variance loss for training the Julep kernel:

$$\mathcal{L}_{\text{Julep}} = \mathbb{V}[W] = \mathbb{V}[U_0(X_0) - U_1(X_1) + \sum \log(k^+/k^-)], \tag{22}$$

where the variance is evaluated over $q_0$ and the law of the forward SDE.

**Flexible neural parametrization.** We stress that any arbitrary parametrization of $A(X,t), b(X,t)$ are all fair game, and the parametrization of $U(X,t)$ also does not break the mathematical framework as long as $U_0$ and $U_1$ stay invariant, which can be easily parametrized as $\widetilde{U}_t = (1-t)U_0 + tU_1 + t(1-t)U_t$, with a free-form $U_t$, which is significantly more flexible than pre-defined linear mixing schedule [36]. When it comes to the noise schedule, although it is possible to prescribe a state-heteroschedastic noise $\epsilon(X, t)$, doing so would require a divergence correction term $\Delta_X \epsilon$ for both the forward and backward SDE. We therefore only optimize $\epsilon$ as a function of $t$.

**Preserving the orthogonal symmetry.** With the amount of care taken to design an orthogonally invariant prior, we cannot afford to lose the $O(n)$-symmetry in the integration stage. Fortunately, this is almost trivial thanks to the rich literature about designing $E(n)$-equivariant force fields and generative models [3–10]—in a sense, we are merely building a *time-dependent* version of these models. Consider such a model $f_\theta : \mathcal{X} \times \mathcal{H} \to \mathcal{X} \times \mathcal{H}$ that map from and to the joint spaces of ($n$-dimensional) geometry $\mathcal{X} \in \mathbb{R}^n$ and semantic embedding $\mathcal{H} \in \mathbb{R}^C$ such that it is permutationally, rotationally, translationally, and reflectionally equivariant on $\mathcal{X}$ and invariant on $\mathcal{H}$, i.e., $\mathbf{x} \in \mathcal{X}, h \in \mathcal{H}$ and $T : \mathcal{X} \to \mathcal{X}$ is rotation, translation, and reflection, we have:

$$\mathbf{x}_f, h_f = f_\theta(\mathbf{x}, h) \iff T(\mathbf{x}_f), h_f = f_\theta(T(\mathbf{x}), h). \tag{23}$$

We can make this model $O(n)$-equivariant (and time-dependent) by constructing an embedding combining the radial component of $X$, the time representation: $h = [t : ||X||]$. The output of this model is connected to Equation 19 as:

$$b_t = \mathbf{x}_f; U_t = \sum h_{f_0}; A = h_{f_1} h_{f_1}^\top, \tag{24}$$

where the control term reuses the equivariant output directly; the potential term aggregates invariant embeddings among the particles (ubiquitous in force field constructions); and the projection term are parametrized using low-rank form, where $h_{f_{0,1}}$ are channels of the invariant output $h_f$. Note that, the resulting projection term $A$ is on the space of $\mathbb{R}^{N \times N}$, similar to that in graph diffusion [43], so we only linearly combine the positions of the particles without introducing internal rotation, thus easily preserving orthogonal symmetry. Although most of these architectures can be reduced to linear complexity, they require pre-specified graph structure (edge connection), which is not possible here. So this backbone also incurs a $\mathcal{O}(N^2)$ runtime complexity. In this paper, we used the simplest equivariant graph neural networks (EGNNs) [3] as the backbone $f_\theta$, and leave more sophisticated architecture for future studies.

**Offline training with replay.** Since the log-variance objective (Equation 22) is an offline objective and can be taken w.r.t. any arbitrary reference path measure, we can simultaneously reduce the variance of the objective and speed up the training by conducting multiple gradient descend step with one sampled trajectory. Specifically, during training, we use one set of the parameters of the kernel $\theta_k$ to sample one trajectory $X_{0 \leq t \leq 1, \theta_k}$. For the next $R$ steps, while we evaluate the log-variance objective and update its parameters to $\hat{\theta}_k$, we keep $X_{0 \leq t \leq 1, \theta_k}$ unchanged. We tune $R$ as a hyperparameter and fixed $R = 5$ throughout this paper.

## 4 MINTJULEP RESULTS: SEPARATING OPTIMIZATION AND SAMPLING.

Having defined the two components, we now put them in a coherent framework and provide a straightforward recipe to optimize them sequentially:

---

**Algorithm 1:** MintJulep training.

---

**Input:** Energy function $U$.
**Input:** Randomly initialized Mint prior $q$ and Julep kernel $k^+$.
**Input:** Hyperparameters: Integration steps $T$, sample size $S$, and replay time $R$
**Output:** Samples from the Boltzmann distribution (Equation 1)

. . . . . . . . . . . . . . . . . . . . . . . . . . . . . . . . . . . . . . . . . . . . . . . . . . . . . . . . . . . . . . . . . . . . . . . . . . . . . . . . . . . . . .

**while** $\mathcal{L}_{\text{Mint}}(\cdot; q)$ *not converging* **do**
    **for** $i \sim \{1, \ldots, S\}$ **do**
        sample $X_i \sim q$ (Equation 15;
    **end**
    descent $\mathcal{L}_{\text{Mint}}(X; q)$(Equation 17) to optimize $q$
**end**

. . . . . . . . . . . . . . . . . . . . . . . . . . . . . . . . . . . . . . . . . . . . . . . . . . . . . . . . . . . . . . . . . . . . . . . . . . . . . . . . . . . . . .

**while** $\mathcal{L}_{\text{Julep}}(\cdot; k^+)$ *not converging* **do**
    **for** $i \in \{1, \ldots, S\}$ **do**
        sample $X \sim q$ (Equation 15;
        sample $t_i \sim U(0, 1), i = 1, \ldots, T$ and sort;
        **for** $i \in \{1, \ldots, T-1\}$ **do**
            sample $X_{t_{i+1}} \sim k^+(\cdot | X_{t_i})$ (Equation 19)
        **end**
    **end**
    **for** $i \in \{1, \ldots, R\}$ **do**
        descent $\mathcal{L}_{\text{Julep}}(\overline{\{X_t\}})$(Equation 22) to optimize $k^+$
    **end**
**end**

---

Note that two stages are required in the training of the model. In the first stage, the Mint prior $q$ is optimized to descend the surrogate KL divergence between $q$ and $p \propto \exp(-U)$, so that it becomes already close to $p$. Next, the details of the distribution, which cannot be filled by elliptical probability mass, are refined in the second stage using the Julep kernel. Although it is also possible to optimize the parameter using the log-variance loss (Equation 22), we found that doing so harms the stability of the training process. Empirically, the training of the prior, due to the lack of SDE integration, only takes seconds on a GPU to converge.

Next, we test the performance of this formulation using synthesized and real-world energy landscape. Again, it is worth emphasizing that we only have access to the target energy function, or probability density up to a constant, and do not have access to samples, which explains the seemingly slightly inferior numerical performance to methods which do require such access [52]—these are two different settings that are not comparable.

**Synthesized energy landscape** We first turn our attention to the time-tested synthetic energy functions defined pairwise distance among particles—Leonard Jones (LJ, Appendix Equation 41) and double wall (DW, Appendix Equation 40). Evidently, the LJ potential increases rapidly when $r \to 0$, which physically represents the strong repulsion among particles when they are about to collide,

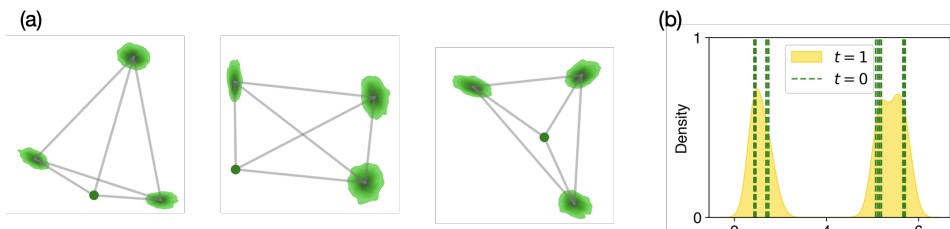

Figure 2: **Mint minimizes; Julep samples**—Illustration of the DW4 experiment. (a) Modes discovered by Mint on the 2-dimensional space. The dot represents the *anchor atom*. Kernel density estimation (KDE) plots from samples taken from the prior. The probability masses, though constrained to be elliptical in shape, are already placed at the minima of the energy surface. (b) KDE plots of the distances among particles from the posterior $t = 1$, with modes of the prior $t = 0$ marked by verticle lines.

|  | DW-4 | LJ-13 | LJ-55 |
|---|---|---|---|
| PIS [24] | $46.2 \pm 8.1$ | $1.2 \pm 1.1$ | $0.1 \pm 0.0$ |
| DDS [33] | $46.1 \pm 7.6$ | $1.0 \pm 1.1$ | $0.1 \pm 0.0$ |
| Mint only | $52.0 \pm 0.2$ | $3.0 \pm 0.1$ | $0.1 \pm 0.0$ |
| Julep only | $48.4 \pm 0.1$ | $1.2 \pm 0.1$ | $0.1 \pm 0.0$ |
| MintJulep | $92.9 \pm 0.5$ | $47.2 \pm 1.1$ | $1.0 \pm 0.6$ |

Table 1: **MintJulep efficiently samples energy functions.** Effective sample size (ESS, %) normalized by the total sample size compared with state-of-the-art path-based models.

which contributes significantly to the *ruggedness* of the energy landscape. This poses a significant challenge for numerical optimization—when a linear path is used, the gradient soon causes overflow because of the 12-th power term. We therefore adopt a smooth annealing path:

$$\tilde{r} = r + \sigma(1 - t); \tilde{U}_{\text{LJ}} = \epsilon/\tau[(\tilde{r}/\sigma)^{-6} - (2 - t)^{-6}]^2, \tag{25}$$

which preserves the minima but slowly anneal the minimal distance among particles from $\sigma$ to $0$ as $t : 0 \to 1$. We reuse this annealing path in the real-world experiment as well.

We compare the sampling efficiency (noted by the effective sample size, ESS, Appendix Equation 42) with the path integral sampler (PIS) [24], which is an online model that propagates the gradient across the SDE, and the denoising diffusion sampler (DDS) [25], which proposes the log-variance objective. Both of these methods use an isotropic Gaussian distribution and an additive kernel, which might explain the drastic difference in the performance. In Figure 2, we show the compartimentalization and collaboration of the two parts of the model, with minima discovered firstly by Mint and refined by Julep. This trend is repeated in Figure 3 as well. Finally, we also conduct an ablation study where we only conduct one improvement at a step, to test the individual ability of the Mint prior and the Julep kernel. Overall, Mint provides more significant improvement across all tasks (compared to the baseline DDS method), but more drastic improvement is only observed when they work in tandem—the ten-fold ESS improvement on the most challenging task of LJ-55 is only observed when both methods were employed.

**Real-world energy landscape: alanine dipeptide (AA).** Having established the satisfactory performance on synthetic sandboxes, we move on to test if our model can achieve real world utility by accelerating molecular dynamics (MD) simulation of biomolecular systems. In such case, the energy function comes from a molecular mechanics (MM) force field (Appendix Equation 42); for a machine learning community-friendly explanation, see [53]. In our setting, we adopt the topology from an alanine dipeptide and uses the collection of parameters from [54]. The protocols of the experiments are detailed in the Appendix.

Reusing the annealing path (Equation 41), we notice a similar trend as the toy experiment—Mint captures the location of the minima quickly while Julep completes the fine detail of the energy landscape. It is worth noting that, since the chirality, which is an important trait of the biomolecules,

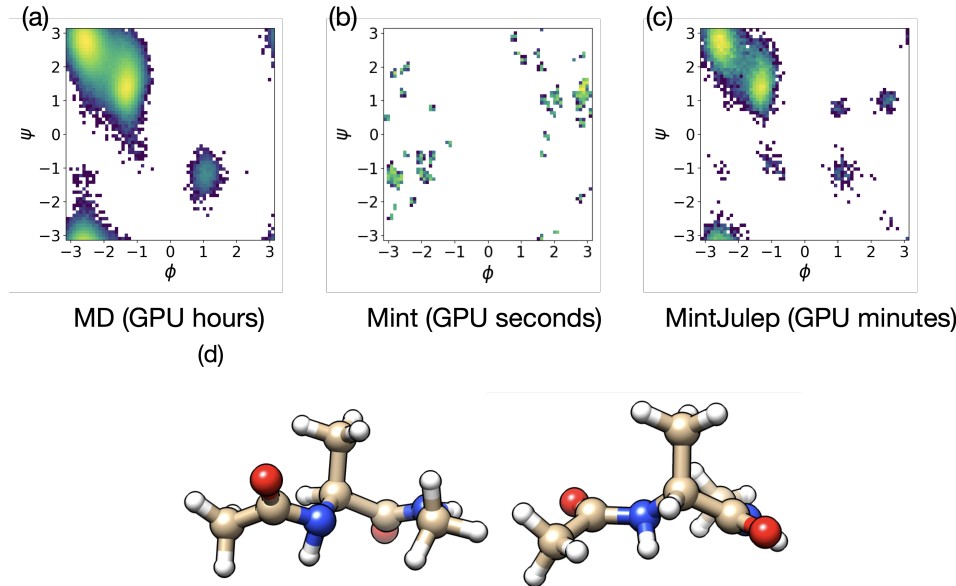

Figure 3: **MintJulep can be used to produce convergent molecular dynamics (MD) simulation trajectories.** Ramachandran plot (KDE plot of the dihedral angles of the molecule) of: (a) Reference equilibrium MD trajecotries of alanine dipeptide; (b) Samples generated from Mint, which captures the minima; (c) Samples generated from MintJulep, which contains finer detail. (d) Representative samples from MintJulep. Note that both chirality is possible since it is not specified in the energy function. The path ESS for MintJulep is $18.0 \pm 1.2\%$.

is not encoded in the energy function, the model may generate samples that has different chirality than those abundant in nature (Figure 3 (d)), which might also explain the additional minima on the Ramachandran plot. In sum, a very brief Mint training can already capture the minima of the energy landscape of alanine dipeptide, whereas MintJulep can recover most of the regions sampled by GPU-days-worth of MD trajectory in less than an hour of training.

## 5 CONCLUSION

If we were able to sample the Boltzmann distribution associated with various physical systems efficiently and accurately, we would be able to build a more reliable bridge between the microscopic and the macroscopic, with which we can gain a deeper quantitative understanding of such systems, thereby rationally designing better pharmaceuticals, materials, and other physicochemical entities with microscopic structure and macroscopic functions. The approach presented here, MintJulep, represents a meaningful step towards this goal.

Concretely, the Boltzmann distributions associated with physical systems can oftentimes be described as *rugged*, i.e. with isolated minima. Another feature of such functions is that they are almost always $O(n)-$invariant. Starting from these two features of the realistic systems, as well as the failure mode of traditional neural samplers [55], we design a brand new class of distributions (Mint) and a powerful discretized kernel associated with the non-equilibrium annealing dynamics. These two practical improvements drastically increase the performance of the path-based neural samplers, allowing us to rapidly generate samples from the Boltzmann distributions associated with real systems given only the energy function. Informally, the Mint prior can be viewed as a minimization step (albeit still preserving the entropy structure), respecting the multimodality of the energy landscape with the mixture of component design. As such, the prior is close to the desired target energy function by KL divergence, leaving the training of the already powerful Julep kernel a breeze.

**Limitations.** As discussed in §2, 3, both Mint and Julep incur $\mathcal{O}(N^2)$ runtime complexity, and need further speed up before they can be efficiently used on realistic protein systems containing

thousands of atoms. Furthermore, the reduction from the $E(n)$ to $O(n)$ group requires a careful choice of *anchor atom*, and the representation power of the internal coordinate system is sensitive to such choices.

**Future directions.**    We plan to investigate further methods to simplify and accelerate the optimization of the Mint prior and the integration of the Julep kernel. This would allow us to model larger protein systems, and unify the pipelines of docking, sampling, and folding within one method. In addition, to make this model generalizable, in the style of [32], is a natural next step.

**Ethics statement.**    We acknowledge and adhere to the Ethnics statement of the ICLR.

**Reproducibility statement.**    The implementation of our method can be found at `https://anonymous.4open.science/r/mint_julep-22D7/`. Interestingly, our method only requires an energy function and is therefore a *data-free* method, requiring no datasets or data-processing.

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

**Proof of Remark 1.1**

*Proof.* For a single kernel $k_t$ of the stated form, consider two input laws $\mu, \nu$ and let

$$X_0 \sim \mu, \quad Y_0 \sim \nu$$

be an optimal coupling for $W_1(\mu, \nu)$, so that

$$\mathbb{E}\|X_0 - Y_0\| = W_1(\mu, \nu).$$

Let $Z \sim \mathcal{N}(0, 2\varepsilon I)$ be independent of $(X_0, Y_0)$ and define

$$X_1 = f_t(X_0) + Z, \qquad Y_1 = f_t(Y_0) + Z.$$

Then $\mathrm{Law}(X_1) = \mu k_t$ and $\mathrm{Law}(Y_1) = \nu k_t$, and by construction

$$X_1 - Y_1 = f_t(X_0) - f_t(Y_0).$$

Using the $L$-Lipschitz property of $f_t$,

$$\mathbb{E}\|X_1 - Y_1\| = \mathbb{E}\|f_t(X_0) - f_t(Y_0)\| \le L \, \mathbb{E}\|X_0 - Y_0\| = L \, W_1(\mu, \nu).$$

Since $W_1(\mu k_t, \nu k_t)$ is the infimum of $\mathbb{E}\|X_1 - Y_1\|$ over all couplings, this gives

$$W_1(\mu k_t, \nu k_t) \le L \, W_1(\mu, \nu).$$

Thus each $k_t$ is $L$-Lipschitz w.r.t. $W_1$.

For the composition $k_{0:T-1} := k_{T-1} \circ \cdots \circ k_0$, iterating the above bound yields

$$W_1(\mu k_{0:T-1}, \nu k_{0:T-1}) \ \le \ L^T \, W_1(\mu, \nu).$$

Now apply this with $\mu = q_0$ and $\nu = \mu$ (the reference measure). By assumption $W_1(q_0, \mu) \le 1$, and

$$q_T = q_0 k_{0:T-1} = q_1, \qquad \mu_T := \mu k_{0:T-1}.$$

Then

$$W_1(q_1, \mu_T) = W_1(q_0 k_{0:T-1}, \mu k_{0:T-1}) \le L^T W_1(q_0, \mu) \le L^T.$$

Finally, by the triangle inequality,

$$W_1(q_1, \mu) \le W_1(q_1, \mu_T) + W_1(\mu_T, \mu) \le 2L^T,$$

so up to an inessential constant (absorbed into the normalization of $W_1$), the requirement $W_1(q_1, \mu) = W_1$ implies

$$W_1 \le L^T \quad \Rightarrow \quad T \ge \log_L W_1.$$

$\square$

**Proof of Theorem 2.6**

*Proof.* Suppose we have a probability density function $p$ that is orthogonally invariant according to Definition 2.1. It can be written as:

$$p(X) = \int \mathrm{d}Y \, \delta(X - Y), \tag{26}$$

which, since $p$ is piecewise continuous, can be approximated arbitrarily well by a Riemann sum:

$$p(X) = \frac{1}{k} \sum k_i(X|\xi_i), \tag{27}$$

where $\xi_i$ is a region in which $k_i$ stays constant. Due to the first fundamental theorem of the orthogonal group, $\xi$ can be embedded in any coordinate system up to the orthogonal transformation. As such, the mixture component

$$k_i \mathrm{PvMFLogNormal}(\cdot, \mu = ||\Xi||, \kappa \to \inf, \rho = ||\Xi||, \sigma \to 0), \tag{28}$$

where $\Xi Q \in \xi, \forall Q Q^\top = I$, can approximate any region $\xi$ arbitrarily well. $\square$

**Proof of Remark 3.2** First, we denote the upper bound of the L2 discretization error by

$$D(t) = \sup_{0 \le s \le t} \mathbb{E}|X_s - \hat{X}_s|^2. \tag{29}$$

Let us also denote the nearest discretization point less thant $t$ as $\tau_t$,

$$D(t) = \sup_{0 \le s \le t} \mathbb{E}|\int_0^{\tau_s} \mathrm{d}u[b(X_u) - b(\hat{X}_u)] + \int_0^{\tau_s} \mathrm{d}W[\sigma(X_u) - \sigma(\hat{X}_u)] \tag{30}$$

$$+ \int_0^{\tau_s} \mathrm{d}u[\exp A(X_u)] - \tau_s \exp A(\hat{X}_u) + \int_{\tau_s}^1 \mathrm{d}u[b(X_u) + \exp A(X_u)] + \int_{\tau_s}^1 \mathrm{d}W\sigma(X_u)| \tag{31}$$

$$\le 5 \sup_{0 \le s \le t} \mathbb{E}|\int_0^{\tau_s} \mathrm{d}u b(X_u) - b(\hat{X}_u)|^2 + |\int_0^{\tau_s} \mathrm{d}W\sigma(X_u) - \sigma(\hat{X}_u)|^2 \tag{32}$$

$$+ |\int_0^{\tau_s} \mathrm{d}u[\exp A(X_u)] - \tau_s \exp A(\hat{X}_u)|^2 + |\int_{\tau_s}^1 \mathrm{d}u[b(X_u) + \exp A(X_u)]|^2 + |\int_{\tau_s}^1 \mathrm{d}W\sigma(X_u)|^2 \tag{33}$$

$$\le 5 \sup_{0 \le s \le t} \mathbb{E} \int_0^{\tau_s} [\mathrm{d}u b(X_u) - b(\hat{X}_u)]^2 + \mathbb{E} \int_0^{\tau_s} \mathrm{d}W[\sigma(X_u) - \sigma(\hat{X}_u)]^2 + \mathbb{E} \int_0^{\tau_s} \mathrm{d}u[\exp A(X_u)]^2 + \tau_s^2 \exp(A(\hat{X}_u))^2 \tag{34}$$

$$+ \mathbb{E} \int_{\tau_s}^1 \Delta t \mathrm{d}u[b(X_u) + \exp A(X_u)]^2 + \mathbb{E} \int_{\tau s}^1 \mathrm{d}u|\sigma(X_u)|^2 \tag{35}$$

$$\le 5 \sup_{0 \le s \le t} 2L^2 \mathbb{E} \int_0^{\tau_s} \mathrm{d}u|X_u - \hat{X}_u|] + 2L^2(\Delta t + 1)\mathbb{E} \int_{\tau s}^1 \mathrm{d}u(1 + |X_u|^2) \tag{36}$$

$$\le 5(2L^2 \int_0^{\tau_s} \mathrm{d}u D(u) + 2L^2(\Delta t^2 + \Delta t + 1)(1 + \sup_{0 \le t \le 1} \mathbb{E}|X_t|^2)), \tag{37}$$

where Cauchy–Schwarz's inequality was used in the first two inequalities, Itô's isometry used in the second inequality, and the Lipschitz and linear growth condition used in the final relation. This can be written as:

$$D(t) \le C(\int_0^t \mathrm{d}u D(t) + \Delta t) \tag{38}$$

Applying Grönwall's inequality, we arrive at:

$$D(t) \le C\Delta t, \tag{39}$$

which recovers Remark 3.2 after taking the square root on both sides and applying Jensen's inequality.

**Energy functions used as targets.** The synthetic energy functions studied in this paper are based upon the pairwise distance $r_{ij} = ||\mathbf{x}_i - \mathbf{x}_j||$. Specific cases include double wall (DW):

$$U_{\mathrm{DW}}(r) = \frac{1}{\tau}[\lambda_2(r - r_0)^2 + \lambda_4(r - r_0)^4], \tag{40}$$

with $\lambda_2 = -4, \lambda_4 = 0.9, \tau = 1$, and Leonard-Jones (LJ) [56] potential:

$$U_{\mathrm{LJ}}(r) = \frac{\epsilon}{\tau}[(r/\sigma)^{12} - (r/\sigma)^6], \tag{41}$$

with $\sigma = 1, \tau = 1$, and an additional harmonic potential constraining particles to the center-of-mass added [52] to prevent the dissolution of the system.

For real-world systems, we consider the molecular mechanics (MM) force field, typically expressed as:

$$
\begin{aligned}
U_{\text{MM}}(\mathbf{x}; \Phi_{\text{FF}}) \quad &= \sum_{\text{bond}} \quad \frac{K_r}{2}(r_{ij} - r_0)^2 \\
&+ \sum_{\text{angle}} \quad \frac{K_\theta}{2}(\theta_{ijk} - \theta_0)^2 \\
&+ \sum_{\text{torsion}} \quad \sum_{n=1}^{n_{\max}} K_{\phi,n}\left[1 + \cos(n\phi_{ijkl} - \phi_0)\right] \\
&+ \sum_{\text{Coulomb}} \quad \frac{1}{4\pi\epsilon_0}\frac{q_i\, q_j}{r_{ij}} \\
&+ \sum_{\text{LJ}} \quad 4\epsilon_{ij}\left[\left(\frac{\sigma_{ij}}{r_{ij}}\right)^{12} - \left(\frac{\sigma_{ij}}{r_{ij}}\right)^6\right],
\end{aligned}
$$

where the total potential energy $U_{\text{MM}}$ as a function of the coordinates of the system $\mathbf{x}$ and the collection of force field parameters $\Phi_{\text{FF}} = \{K_r, K_\theta, r_0, \theta_0, K_{\phi,n}, \phi_0, q, \sigma, \epsilon\}_i$ is modeled as the sum of bond, angle, torsion, and nonbonded energy.

**Sampling efficiency metric.** In this paper, we are primarily concerned about the sampling efficiency, characterized by the (normalized) effective sample size (ESS):

$$
\text{ESS} = \frac{1}{n}\mathbb{E}^{-1}[W] \approx \frac{1}{n}\frac{(\sum W_i)^2}{\sum W_i^2} = \frac{1}{n}\sum(\text{SoftMax}^2(W_i))^{-1}. \tag{42}
$$

**Experimental details.** The architectures are implemented in JAX [57] and its eco-system. The random seed is fixed as 2666 everywhere in this paper. We use a 3-layer EGNN [3] with 64-units each and TanH activation everywhere in this paper. The Adam [58] optimizer with learning rate $1e-3$ and L2 regularization $1e-5$ was used. These choices are optimized based on simple experiments on the LJ-13 system. During training, we fix the number of integration step to be $1e2$ and the batch size to be $1e2$. For evaluation, $1e5$ samples are used to compute the ESS.

