# OpenReview forum: "On the practicality of Boltzmann neural samplers"
_ICLR.cc/2026/Conference — Submitted to ICLR 2026_

### Official Review · Reviewer_yXPN · 2025-10-17

**Soundness:** 3
**Presentation:** 2
**Contribution:** 3
**Rating:** 4
**Confidence:** 4

**Summary:**

This work proposes two new techniques to improve neural samplers. The first technique (MINT) is a more expressive initial distribution (prior). These expressive priors are constructed via mixtures of pairwise Fisher-von-Mises distribution (introduced in the work). They are optimized at the first training phase using the REINFORCE algorithm to give a simple yet more expressive “initial” guess of the ground truth distribution. The second technique (JULEP) is a more expressive discrete-time Markov kernel than the Gaussian ones used previously. The JULEP kernel is trained with the same log-variance loss (discrete version) as previous method. Experiments on Lennard-Jones and Alanine Dipetide are presented.

**Strengths:**

- The proposal to use MINT prior is sound and reasonable. The initial phase of minimizations is computationally negligible and makes sense to use for many downstream neural samplers.
- The JULEP kernel is similarly reasonable to use.
- The proposed methods address shortcomings of current neural samplers
- Experiments are shown on both synthetic and a simple molecule

**Weaknesses:**

(1) The writing of the paper can be improved significantly:
   - Motivation and Structure: It is very reasonable to design better priors. However, this motivation is not communicated well in the text. I would recommend stressing this point more in the introduction. Throughout section 2, it is unclear to the reader what role the prior or the class of distributions that is defined plays in the context of neural samplers. Also in the contributions, “A new prior called Mint”. The word “prior” was not used in the text before and it is unclear at this point what “prior” refers to here. One needs to explain what prior is in this context (i.e. the initial distribution of the SDE).
- The “pathology” in remark 1.1. is just postulated and no proof is given or anything. Many other reasons why neural samplers fail could be given and discretization error was so far not one of them, e.g. diffusion models have discretization errors but can express extremely complex distributions with their SDEs.

(2) Some statements have insufficient proofs or no derivations are given for non-trivial statements:
- Theorem 2.6.: It would be to say what “error” means her, i.e. what topology/metric in the space of probabilities are u considering? Further, the proof is not sufficient. Just saying that every region can be approximated arbitrarily does not suffice. More details are needed. Further, the fact that you assume a Riemann sum excludes a set of probability distributions. You need to specify what set of probability distributions you allow for (one that have a density that is Riemann integrable?). Please give a rigorous proof.
- Line 241: “We can see that the constraint in Remark 1.1 no longer holds.” Which constraint? It was not formally defined what that constraint is. It was just postulated. So this statement is not grounded.

(3) The experiments and benchmarks are limited. For example, No numerical benchmarks for alanine dipeptide are presented. Why?

Overall, I found the paper interesting and would be inclined to increase my rating if the above issues are addressed.

**Questions:**

- The kernel in equation (11) does not properly fit to the notation used in previous paragraphs (equation (6)) in particular. It is missing the -epsilon*\nabla U_t term maybe?
- Equation (3): A space (e.g.\quad) after the SDE and before the initialization would be appropriate
- Remark 1.1. Is not really clearly expressed. What does it mean to transport something with 1-Wasserstein distance W_1? Does it mean up to error in W_1? Does it mean that q_1 and q_0 have Wasserstein distance W_1?
- Line 180: r\in N, what is N? Isn’t N an integer?
- Equation (16) requires a derivation. “Recall that” in the sentence before is inappropriate as most readers would not know that.
- Line 241: “We can see that the constraint in Remark 1.1 no longer holds.” Which constraint? It was not formally defined what that constraint is. It was just postulated. So this statement is not grounded.

---

> ### Author Response · Authors · 2025-11-29
> **More motivation and theoretical rigor**
>
> Dear Reviewer `yXPN`,
>
> Many thanks again for your constructive feedback. We have addressed your concerns in our rebuttal, and have introduced more theoretical rigor into our manuscript. Specifically:
>
> > It is very reasonable to design better priors. However, this motivation is not communicated well in the text.
>
> Thank you so much for this suggestion! We have provided more discussions around the motivation of the design of the prior.
>
> > The word “prior” was not used in the text before and it is unclear at this point what “prior” refers to here.
>
> We have included the definition of the prior, i.e. the intial distribution of the SDE when $t=0$, in the manuscript.
>
> > The “pathology” in remark 1.1. is just postulated and no proof is given or anything.
>
> We have included in a proof in the appendix.
>
> > Please give a rigorous proof (for Theorem 1).
>
> Thanks to your suggestions, we have introduced some conditions on the target distribution $p$ to be Riemann-integratable. The proof follows the style of the universal approximation theorem of the vanilla Gaussian mixture. The only difference is that, if the target $p$ is orthogonally invariant, it can be sufficiently described by the dot product used here.
>
> We have also combed through the manuscript and fixed the errors mentioned in your questions section.
>
> Many thanks again for your review!

---

### Official Review · Reviewer_EQvf · 2025-10-19

**Soundness:** 3
**Presentation:** 3
**Contribution:** 2
**Rating:** 4
**Confidence:** 4

**Summary:**

This paper focuses on training neural samplers for sampling n-body system, eg molecules, where we only have access to the target energy or unnormalized density but not samples from the equilibrium distribution. The core of the paper lies in finding a good prior and training a drift function to traverse from this learned prior to the target distribution by minimizing the divergence between path measures. And the main novelty lies in two parts:

1. The author designs a specific prior distribution, instead of the isotropic Gaussian that is usually used, for the O(3) group. This can be seen as analogous to the Mixture of Gaussians, but preserving permutational, rotational, and translational invariance, where the mixture weights and parameters relating to means and covariances are learnable. By minimising the reverse KL of the target and this mixture prior, the author learns a good prior first.

2. To train the drift function, the author follows recent works in neural samplers to optimize the log-variance divergence, while they define a different transition kernel, which sounds novel.

By first learning the prior in (1), and then optimizing the drifts in (2), the author shows that their method could achieve better results than the "classic" neural samplers, PIS and DDS, in both synthetic task (DW-4, LJ-13, and LJ-55) and more practical one (alanine dipeptide on vacuum).

Despite novelty, there are several concerns, which mainly lie in the justification of validility, ablation study, experimental performance, and computational budget. Please see the concerns in the "Weakness" section.

In summary, the reviewer recommends a weak rejection.

**Strengths:**

1. The motivation is clearly written

2. The method provides a contribution for task-specific heuristics of neural samplers for molecules

**Weaknesses:**

## Ablation study

The main weakness of the paper lies on the ablation study. In short, this paper proposes two novel components: the mixture prior and a new transition kernel.

1. To justify the effectiveness of the mixture prior, one should ablate training neural samplers with a mixture-of-Gaussian (MoG) prior with the same number of mixtures and learnable means and covariances (or learnable isotropic variance to reduce the number of parameters). The author could consider training with a MoG prior, ie MoGJulep, to showcase the effectiveness of Mint.

2. To justify the effectiveness of the proposed transition kernel, one should ablate training with the standard Euler-Maruyama (EM) kernel, ie MintEM.

## Justification of the validity of proposed transition kernel

It is not clear why the backward SDE in line 236 is a valid traversal to the target distribution, and also there's lack of mathematical details talking about how the discretized kernel comes and why it holds. Elaborating more details on them in either the main context or the appendix is necessary.

## Sampling from PvMF

Equation (16) shows that PvMF can be asymptomatically approximated by a projected Gaussian, when $\kappa\rightarrow\infty$. However, $\kappa$ can't be infinity: if so, as mentioned in line 166, $\exp(\kappa)$ would be infinity and the integration over a finite-
volume manifold can be infinite. On the other hand, $\kappa$ should be chosen as a finite number in practice, which means the approximation can introduce mismatch. If so, the samples at the beginning of the sampling process are not really prior samples, and therefore the calculation of the path weight can be biased, which amplifies the reviewer's concern on the MoG prior ablation mentioned previously. It would be great that

1. the author could talk more about the hyperparameter setting in practice

2. the author could elaborate more on the mismatch between true PvMF samples and the approximated samples from the projected-Gaussian approximation

## Experimental results

The reviewer mainly concerns on the alanine dipeptide experiments in vacuum.

1. Firstly, it is true that both the energy function and the EGNN can't distinguish chirality, however, the D-form samples can always be transformed to L-form by alignment. Therefore, the author should consider visualizing the Ramachandran plots (figure 2, (b) and (c)) after this D-to-L transformation.

2. On the other hand, the reviewer has a question that, why not also conducting experiments on ALDP in implicit solvent? In fact, this problem is more difficult than the vacuum one. But it would be a good contribution to see if the proposed method could do it.

3. Figure 2 (c) still shows significant difference when comparing to the ground truth. There are extra modes, where the reviewer thinks it is quite not likely due to the chirality, as it would make the Ramachandran plot "mirrored" while not making extra modes. Also, the mode weights are not correct. Therefore, the reviewer recommends using the path weight for importance sampling or resampling and shows how the reweighted/resampled samples look.

**Questions:**

Please see Weakness.

---

> ### Author Response · Authors · 2025-11-29
> **Ablation study and exact sampling**
>
> Dear Reviewer `EQvf`,
>
> Thank you for your thorough and constructive review, according to which we have significantly improved our manuscript.
>
> > The main weakness of the paper lies on the ablation study.
>
> Thank you so much for this comment. We have included an ablation study section in our paper. In Table 1, the _Mint Only_ results correspond to your MintEM suggestion, whereas the _Julep Only_ results employ an isotropic Gaussian prior. Note that a vanilla Gaussian mixture is not orthogonally invariant and is therefore not suitable for a prior in our equivariant setting.
>
> > Sampling from PvMF: On the other hand, $\kappa$ should be chosen as a finite number in practice, which means the approximation can introduce mismatch.
>
> As detailed in the paper, we use a short Langevin dynamics sampling to correct for this mismatch. This, as well as the fact that the distance between the PN distribution and the PvMF distribution is minimal to begin with when $\kappa$ is large (we constrain $\kappa > 1$ in practice), ensures that we are sampling from the true PvMF distribution. Note that the PvMF distribution is orthogonally invariant whereas the approximated PN distribution is not. This is the reason why we have to work with the PvMF distribution during sampling.
>
> > The reviewer mainly concerns on the alanine dipeptide experiments in vacuum.
> Thank you for the suggestions for improving the alanine dipeptide sampling experiment. We have incorporated your suggestions to align D to L conformations. We have also discovered a new trick which replays the trajectory to speed up convergence, with which we are rerunning the experiments and will update the manuscript shortly.
>
> Many thanks for your suggestions!

---

### Official Review · Reviewer_njMA · 2025-10-27

**Soundness:** 1
**Presentation:** 2
**Contribution:** 2
**Rating:** 2
**Confidence:** 4

**Summary:**

This paper introduces two techniques to improve Boltzmann neural samplers. First, it proposed to use a more flexible prior distribution to make the learning of transport easier. Second, it imprvoves the discretisation kernel of the forward and backward SDE to improve the learning of the transports. It denomstrates improvements on systems like DW, LJ and Alanine.

**Strengths:**

This work provides valuable insights into the limitations of current neural samplers by identifying two key challenges and proposing efficient methods to address each of them. The experimental results also demonstrate the effectiveness of the proposed solutions.

Notably, the approach also achieves promising performance on alanine dipeptide in Cartesian coordinates, a particularly challenging benchmark given the absence of training data.

The code is provided.

**Weaknesses:**

While interesting, I have several concerns on this paper, and I do not think it reaches the bar of publication in its current shape. I will explain my reason and try to provide some suggestions:

1. **No Exp Details**: I cannot find any details on the experiments done in this paper. It's fine to drop some details in the main text, but there is also no details in appendix. I can see the code is provided. However, it is important to include details instead of relying on people to read the code.

2. **No Ablation study and  Many design motivation is unclear**: The paper proposes two approaches to improve the training of neural sampler. Also, for the prior, the proposed form is not tractable and requires approaximaton and langevin correction, the loss is a combination of fitting and repulsion. It is important to ablate how these components contributes to the performance.
There are many things unclear in the method. For example,
- How much gain is due to each proposed component?
- why it is proposed to use PvMF instead of its approximation form (the PN in Eq 16)? How much gain will this bring us?
- How the repulsion prevent mode collapsing?
- Is the loss a direct sum of eq 18 or 17? Or should they be weighted? Is the weighting important?
- Why 19 is more flexible? Is this design supported by some evidence?


3. **Writing need to be improved**: The structure and emphasis should be rebalanced. Key design decisions are insufficiently explained, and important components such as the Julep kernel are introduced without justification or intuitive description. In contrast, a substantial portion of the text is dedicated to standard definitions, including the alanine dipeptide energy formulation, LJ, DW, and ESS. This should be shortened or moved to the appendix.
The current presentation gives the impression that some parts of the paper were filled with lengthy formulas due to time constraints rather than to support the main contributions.

4. The title is ambiguous and potentially misleading. The paper focuses on improving the performance of neural samplers, yet the current approach still appears far from being truly practical for real-world Boltzmann sampling.  “On the practicality of Boltzmann neural samplers” sets the expectation that the work either achieves practicality or provides a clear discussion of the remaining gap.
There are also many other challenges for neural samplers, like mode balancing [1] and energy evaluation efficiency [2].

5. Mssing references: [3] learns the prior as well. [4] uses also differrent kernels instead of standard EM discretisation (exp integrator)

[1] Grenioux, L., Noble, M., & Gabrié, M. (2025). Improving the evaluation of samplers on multi-modal targets.

[2] He, Jiajun, et al. "No Trick, No Treat: Pursuits and Challenges Towards Simulation-free Training of Neural Samplers.".

[3] Blessing, Denis, Xiaogang Jia, and Gerhard Neumann. "End-to-end learning of gaussian mixture priors for diffusion sampler." ICLR

[4] Phillips, Angus, et al. "Particle Denoising Diffusion Sampler." ICML.

**Questions:**

I am not sure I understand how eq 19 converges to the formula in Line 236? I may miss the definitation of W_t.

---

> ### Author Response · Authors · 2025-11-29
>
> Dear Reviewer `njMA`,
>
> Thanks so much for your thorough and constructive feedback. We have significantly reworked our paper based on your suggestions.
>
> > No Exp Details
>
> We have included a detailed experimental procedure in the appendix.
>
> > No Ablation study
>
> We have conducted an ablation study and have included it in Table 1.
>
> > Many design motivation is unclear
>
> We have reworked our introduction section and included more discussion around the motivation. Specifically, the reason why we need to use the proposed PvMF rather than the PN distribution as a prior is because PvMF is naturally rotationally invariant, whereas the PN distribution is not, if we do not sample the infinite many random rotations.
>
> > Writing need to be improved: The structure and emphasis should be rebalanced.
>
> Thank you for this suggestion. We have reworked the manuscript to focus more on the technical novelties and have deferred the details to the appendix.
>
> Thank you so much again for your extremely helpful review!

---

### Official Review · Reviewer_ivNK · 2025-11-03

**Soundness:** 2
**Presentation:** 1
**Contribution:** 2
**Rating:** 2
**Confidence:** 3

**Summary:**

This manuscript introduces a new framework for defining and training neural generators, which the authors claim significantly enhances the practical performance of such samplers when exploring rugged energy landscapes. The proposed approach combines two main innovations: (i) the introduction of a flexible prior capable of capturing mixtures of densities of states while preserving the system’s symmetries, and (ii) the incorporation of a discretized kernel designed to increase the expressiveness of the model. The authors also present a limited set of numerical experiments that illustrate the potential of the method.

**Strengths:**

The introduction of the paper is well organized, beginning with a clear discussion of the limitations of current normalizing flow approaches for accelerating molecular dynamics simulations, followed by a well-motivated hypothesis explaining why these methods may fail. This serves as an effective introduction to the authors’ proposed solution. The problem addressed by this new framework is highly relevant, and the proposed strategy to enhance the expressiveness of these models could prove to be particularly valuable.

**Weaknesses:**

First, the paper is very challenging to follow. The introduction of the new framework is presented in an extremely technical manner, making it difficult to understand for readers in the computational physics or chemistry communities, which are the primary audience of this work. As a result, the details of the proposed method are hard to grasp. The manuscript would benefit considerably from moving many of the technical derivations to an appendix, complemented by a more intuitive explanation and a schematic illustration of the method in the main text. This would free space to better demonstrate the practical implications and performance of the proposed approach.
Second, the experimental section is clearly insufficient. The authors neither benchmark their method against established approaches nor test it on sufficiently challenging problems. The only comparison provided concerns the performance of different annealing trajectories in Table 1. However, the success of annealing trajectories strongly depends on avoiding first-order phase transitions during the annealing process, a factor that is highly problem- and trajectory-dependent. Consequently, this comparison is not particularly informative.
Figure 2 illustrates that the method can identify most modes of a target distribution when compared to molecular dynamics simulations. Nonetheless, the manuscript lacks a discussion on whether the generated proposals would be accepted as valid moves in actual molecular dynamics or Monte Carlo simulations, which is essential to assess their potential for accelerating equilibrium sampling.
Overall, to be suitable for publication, the paper should include more substantial experimental validation and quantitative comparisons with existing methods. In addition, a discussion of the method’s limitations—particularly regarding computational cost and the impact of the choice of the anchor atom—should be incorporated into the main text.

**Questions:**

1. How does the proposed framework compare quantitatively with existing methods? A direct performance comparison—both in terms of sampling efficiency and computational cost—would greatly help to assess the practical relevance of the approach.

    2. Can the authors quantify the acceptance rate of the proposed moves when evaluated through a Metropolis test? This would provide a concrete measure of whether their generated proposals are suitable for accelerating equilibrium sampling.

    3. The presentation of the method is currently very technical and difficult to follow for readers outside the immediate field. Could the authors make the explanation more accessible, for instance by providing an intuitive overview and a schematic representation of the algorithm before introducing the full mathematical formalism?

---

### Author Response · Authors · 2025-11-28
**New findings, more theoretical and experimental details, ablation study, and motivation**

Dear reviewers, thank you all for your constructive feedback, based on which we have significantly improved our manuscript. In this thread, we would like to first provide a general response to the common questions and feedback provided by you all. Individual questions are further addressed in separate posts. Please also see our updated manuscript with the changes highlighted.

## Recap of contributions
In this paper, we tackle a challenging problem in the computational modeling of physical systems, namely to sample the Boltzmann distribution given only its energy function. We show that traditional methods, though elegant in theory, are practically not as expressive due to the inability of discretized kernels to bridge distant distributions. To precisely overcome this limitation, we propose two improvements to the existing framework—
1. A new class of prior (initial tractable distribution of the SDE), Mint, which fits the energy landscape roughly without integration.
2. A new, multiplicative, non-local discretized kernel, Julep, which bridges distant distributions.
We theoretically show and experimentally verify the utility of each of these improvements.

## New findings during rebuttal
We have continuously improved our manuscript during the rebuttal period, and introduced new findings. We have found that employing a replay buffer and reusing the trajectories to conduct multiple rounds of gradient updates will drastically improve the performance of the model while also reducing the cost. We have updated the numerical experiment results accordingly.

## More theoretical and experimental details
Thank you, Reviewers `njMA` and `yxPN`, for suggesting more details in the experimental and theoretical work, respectively. In the appendix, we have provided more thorough experimental details as well as more detailed proofs.

## Ablation study
We would like to thank the reviewers again for recommending an ablation study. We have conducted one incorporating only the Mint prior and only the Julep kernel, respectively. We observe a general trend that the Mint prior contributes more significantly to the performance of the model across all tasks.

## Design motivation
We have reworked our Remark 1.1 and the discussion around it to further elucidate the pathology of neural samplers, and how a more flexible prior and a non-local kernel can overcome this issue. We have also included a new semantic illustration figure.

---

### Meta-Review · Area_Chair_oNvU · 2026-01-05

**Summary:**

Reviewers' concerns:

1. The paper is challenging to follow (ivNK, njMA, yXPN).
2. The experiments lack baselines (ivNK, njMA) and are not extensive enough to (ivNK, yXPN).
3. Would the moves be accepted (ivNK) / is the Julep kernel valid since it may be based on biased weights (EQvf)?
4. The discretization pathology seems to be novel but is not justified enough, what is the reason given that other systems do not exhibit it? (yXPN)
5. Ablations on the imprtance of the two components of the framwork are missing (njMA).

**Reviewer Concerns:**

The authors overall chose not to address several of the reviewers' concerns. The answers they did give were sparse and largely did not provide the details necessary to consider the concerns addressed.

1. Several changes were implemented to the writing, it is unclear whether the changes requested by reviewer ivNK were implemented since the authors chose not to answer that reviewer.
2. No baselines were added. The reviewers mentioned as missing: Blessing et al. ICLR 2025 and Philips et al. ICML 2025, and the authors did not comment on whether these baselines were necessary.
3. Not addressed by the authors.
4. A proof was given for Remark 1.1, but the authors did not give further details on how this realizes in practice.
5. The authors did provide an ablation on leaving out one of the two major components of their framework.

**Reviewer Scores:**

Since the authors chose not to answer most concerns in detail, I do not think that the reviewers would have increased their scores.

---

### Decision · Program_Chairs · 2026-01-26

Reject